# Genomic landscape of pathogenic mutation of *APC*, *KRAS*, *TP53*, *PIK3CA*, and *MLH1* in Indonesian colorectal cancer

**Vania Myralda Giamour Marbun**[1]*, **Linda Erlina**[2], **Toar Jean Maurice Lalisang**[1]

**1** Department of General Surgery, Digestive Division, Faculty of Medicine Universitas Indonesia, Jakarta, Indonesia, **2** Department of Medical Chemistry, Faculty of Medicine Universitas Indonesia, Jakarta, Indonesia

☯ These authors contributed equally to this work.

* vania.myralda@gmail.com

**Data Availability Statement:** The data underlying the results presented in the study are available from Figshare. Marbun, Vania Myralda (2021):

## Abstract

### Background

Colorectal cancer (CRC) needs several mutations to occur in various genes, and can vary widely in different individuals; hence it is essential to be discovered in a specific population. Until recently, there has been no known study describing *APC*, *TP53*, *PIK3CA*, *KRAS*, and *MLH1* of CRC in Indonesian population. This study describes the nature and location of mutation in CRC patients treated at three different hospitals in Jakarta.

### Methods

This descriptive study was conducted on CRC patients who underwent neoadjuvant, surgical, and adjuvant therapy at RSCM, RSKJ, and MRCCC in 2017–2018. DNA analysis was performed using next-generation sequencing and aligned against GRCh38. The pathogenic variant was identified using ACMG classification and FATHMM score. Data related to behavior and survival were collected from medical records.

### Results

Twenty-two subjects in which *APC*, *TP53*, and *PIKCA* were mutated. *KRAS* mutation occurred in 64%, while *MLH1* in 45%. There were five mutation types: nonsense, missense, frameshift, splice-site, and silent mutation. There are four groups of co-occurring mutations: *APC*, *TP53*, *PIK3CA* (triple mutation/TM) alone; TM+*KRAS*; TM+*MLH1*; and TM+*KRAS*+*MLH1*, presenting different nature and survival.

### Conclusion

Indonesia has a distinct profile of pathogenic mutation, mainly presenting with locally-advanced stage with various outcomes and survival rate.

mutasi gen KKR indonesia 100621.xlsm. figshare. Dataset. https://doi.org/10.6084/m9.figshare.16656808.v1.

**Funding:** The author received no specific funding for this work. PUTI Q1 2020 from Universitas Indonesia with Grant number: NKB-1312/UN2.RST/HKP.05.00/2020.

**Competing interests:** The authors have declared that no competing interests exist.

**Abbreviations:** ACF, aberrant crypt foci; ACMG, American College of Medical Genetics; ACS, adenoma-carcinoma sequence; AJCC, American Joint Committee on Cancer; *APC*, Adenomatous Polyposis Coli; *BAX*, Bcl-2 Associated X protein; *CDC4*, Cell division control protein 4; CRC, colorectal cancer; DCC, deleted in colorectal cancer; DNA, deoxyribonucleic acid; *EGFR*, Epidermal Growth Factor Receptor; FAP, Familial Adenomatous Polyposis; FATHMM, Functional Analysis through Hidden Markov Models; FFPE, Formalin-fixed Paraffin-Embedded; GRCh38, *Genome Reference Consortium Human Reference 38*; IGF2R, Insulin Like Growth Factor 2 Receptor; *KRAS*, Kirsten rat sarcoma viral oncogene; MFOLFOX6, modified Folinic Acid + 5-Fluorouracil + Oxaliplatin; *MLH1*, MutL Homolog 1; MMR, mismatch repair; *PIK3CA*, Phosphatidylinositol-4,5-Bisphosphate 3-Kinase Catalytic Subunit Alpha; *SMAD4*, SMAD family member 4, Mothers against decapentaplegic homolog 4; SNPEFF, SNP Effect; SNV, Single Nucleotide Variant; *TGFBR2*, Transforming Growth Factor Beta Receptor 2; *TP53*, Tumor Protein 53; VCF, Variant Call Format; *VEGF*, Vascular Endothelial Growth Factor; Wnt, Wingless-related integration site.

# Introduction

Colorectal cancer has been known as one of the most well-studied malignancies. Its dynamic and heterogeneity are characterized by many interconnecting molecular etiopathogeneses exhibiting different behavior inter and intratumor [1–5]. Based on recent biomolecular studies, genetic and epigenetic analysis can evaluate the nature of the tumor, hence, able to predict heredity, progressivity, recurrency, response to therapy, and even survival rate. Those variables cannot be estimated by the AJCC staging system alone. For this reason, precision medicine rooting in the genomic profile of each individual is starting to advance.

Colorectal malignancy, which involves at least three or four genetic mutations, is feasible for next-generation sequencing methods [2, 6]. Two of the three most common carcinogenic pathways are chromosomal and microsatellite instability [7–10]. Five genes which frequently involved are *APC*, *TP53*, *KRAS*, *PIK3CA*, and *MLH1*. Different groups of age, gender, and geographic location have different variations of mutation and genes involved, so that study on a specific population is essential in advancing precision medicine [11]. Until recently, there has been no publication providing the genomic landscape of colorectal cancer in Indonesian population. This study aims to analyze the genomic profile of colorectal cancer in Indonesia.

# Methods

This is a descriptive study in patients with colorectal malignancies who underwent surgery, chemoradiation, chemotherapy at RSCM, RSKJ, and MRCCC in 2017–2018 whose tumor tissue specimens were still stored correctly in the form of formalin-fixed paraffin-embedded (FFPE). This study has been reported in line with STROCSS criteria [12].

## Sample preparation

The Department of Medical Chemistry, Faculty of Medicine, Universitas Indonesia at Bioinformatics Core Facility of Indonesia Medical Education and Research Institute (IMERI) performed all sequencing preparation.

DNA extraction was performed using the QIAamp DNA FFPE Tissue Kit. The quality of extracted DNA was evaluated using an absorbance ratio of 260 nm to 280 nm ($A_{260}/A_{280}$) and 260 nm to 230 nm ($A_{260}/A_{230}$). The purity criterion for samples with the $A_{260}/A_{280}$ ratio is within the range of 1.8–2.0, and the $A_{260}/A_{230}$ ratio is within 2.0–2.2. After the purity criterion was fulfilled, sequencing was done utilizing *AmpliSeq Cancer HotSpot Panel v2 for Illumina*. Results in FASTQ format were quality-checked with FASTQC (v.0.9.5; http://www.bioinformatics.babraham.ac.uk/projects/fastqc/) and aligned against *Genome Reference Consortium Human Reference* 38 (GRCh38). Variant calling was done using LoFreq, annotated with SNPEFF, and filtered with SNPSift. Annotation results were stored in a variant call format (VCF) file.

## Inclusion criteria, exclusion criteria, and identification of pathogenic mutation

1. Retrieval of VFC files fulfilling inclusion and exclusion criteria

   • Inclusion criteria: FFPE samples fulfilling DNA purity criterion and showing PASS status in FASTQC

   • Exclusion criteria: none

2. Data filtering based on estimation of putative impact or deleteriousness showing "HIGH".

3. Identification of single nucleotide variant (SNV) and synchronization with three databases

- ClinVar (https://www.ncbi.nlm.nih.gov/clinvar/)
- COSMIC (https://cancer.sanger.ac.uk/cosmic)
- The Ensembl project (https://asia.ensembl.org/info/index.html)

4. Identification of somatic effect based on American College of Medical Genetics (ACMG) classification and Functional Analysis through Hidden Markov Models (FATHMM) score.

5. Selection of SNVs meeting pathogenic criteria (ACMG pathogenic variant or FATHMM score ≥ 0,7 or both.

6. Identification of SNVs of *APC*, TP53, PIK3CA, KRAS, and MLH1

7. Matching samples' numbers and medical record data

## Ethical approval

The Ethics Committee of the Faculty of Medicine, Universitas Indonesia–RSUPN Cipto Mangunkusumo regarding the Protection of human rights and welfare in medical research, has carefully reviewed the research with registry number: KET-445/UN2.F1/ETIK/PPM.00.02/2021. All procedures of Ethical Approval are performed in accordance with ICH-GCP standard procedure. All data were fully anonymized and Ethics committee waived the requirement of the informed consent.

## Results

### Patients characteristics

Twenty-two samples were collected in accordance with the sample preparation procedures mentioned above. Among these samples, 41% (9/22) were diagnosed with stage 3b, of which 7 were elective cases. Fifty-nine percent (13/22) had lymphovascular invasion, of which one was diagnosed with stage 2A, and 12 were in stage 3B-4C.

### Pathogenic mutation mapping in whole chromosomes

There were pathogenic mutations in almost all somatic chromosomes except 6, 9, 14, 16, 21, and 22, which involved 25 genes and 641 SNV (Table 1). Three mutation types were identified, i.e., synonymous (silent mutation), nonsynonymous (nonsense, missense, and frameshift), and splice-site mutation.

### Pathogenic mutation mapping of *APC, TP53, PIK3CA, KRAS,* and *MLH1*

Two *APC* pathogenic mutations occurred concurrently (nonsense and missense) in 1 patient. *TP53* also had five coherent mutations in 1 patient (nonsense, missense, frameshift, silent, and splice-site) and only 3 of 22 patients had missense mutation. Only 1 type of pathogenic mutation occurred in *MLH1* (nonsense) and *PIK3CA* (missense). Singular *KRAS* mutation occurred in 10 patients (8 missense and 2 silent), and multiple mutations occurred in 4 patients (Table 2).

Co-occurring mutations in more than three genes were presented in all subjects. A combination of triple mutation (*APC, TP53, PIKCA*) occurred in 4 of 22 patients. A combination of quintuple mutation (*APC, TP53, PIKCA, KRAS, MLH1*) occurred in 6 of 22 patients (Table 3).

**Table 1. Pathogenic mutation mapping.**

| Chr | 1 | 2 | | | 3 | | | | 4 | 5 | 7 | | 8 | 10 | | | 11 | 12 | 13 | 15 | 17 | 18 | 19 | 20 | | Total Genes | Total SNVs |
|---|---|---|---|---|---|---|---|---|---|---|---|---|---|---|---|---|---|---|---|---|---|---|---|---|---|---|---|
| Gene | a | b | c | d | e | f | g | h | i | j | k | l | m | n | o | p | q | r | s | t | u | v | w | x | y | | |
| Pts | | | | | | | | | | | | | | | | | | | | | | | | | | | |
| 1(1) | | | | | 2 | | 2 | | 1 | 1 | | 2 | | | | | | 2 | 1 | | 4 | 5 | 1 | | | 10 | 21 |
| 2(3) | | | | | 1 | 1 | 1 | | 1 | 3 | 1 | | | 1 | 1 | 1 | | | 4 | 1 | 10 | 1 | | 1 | 1 | 15 | 29 |
| 3(6) | 1 | | | | 2 | 1 | | | 2 | 3 | 1 | 1 | | | | | | 1 | | | 7 | | | | | 9 | 19 |
| 4(14) | 1 | 1 | | | | | 1 | | 1 | 3 | | | | | | | | 1 | | | 3 | | 1 | | | 8 | 12 |
| 5(15) | 1 | | 2 | | 1 | | 1 | 1 | | 2 | 1 | | | 1 | | | | 1 | | | 4 | 2 | | | | 11 | 17 |
| 6(16) | | | | | 1 | | 2 | | 2 | 1 | | | 1 | | | | | 1 | | | 4 | 1 | | | | 8 | 13 |
| 7(19+19b) | | | 2 | | | 2 | 2 | | 1 | 4 | 2 | 1 | | | | | 1 | 2 | 4 | 1 | 14 | 2 | 1 | 1 | | 15 | 40 |
| 8(20) | | | | 1 | 2 | 1 | 1 | | 3 | 2 | | | | | | | | 1 | 5 | 1 | 9 | 2 | 1 | | | 12 | 29 |
| 9(22) | 1 | | | | 1 | | 3 | | 2 | 1 | | | | | | | | | | | 9 | 1 | | | 1 | 8 | 19 |
| 10(23) | | | | | 1 | 2 | 2 | | 1 | | 1 | | | | | | | | 4 | | 4 | 1 | | 1 | | 9 | 17 |
| 11(29) | | | 1 | | 1 | 1 | 1 | | | 3 | 2 | 1 | | | 1 | | | 1 | 1 | | 4 | | 1 | 1 | | 12 | 18 |
| 12(34) | 1 | 1 | | | 2 | 1 | 2 | | | 6 | 1 | 2 | 1 | 1 | | 1 | 1 | | 3 | | 10 | 1 | 1 | | 1 | 17 | 36 |
| 13(3737) | 1 | 1 | | | | 1 | 2 | | 2 | 5 | 2 | 2 | | 2 | | | | 1 | 3 | | 10 | 4 | | 1 | 1 | 15 | 38 |
| 14(9) | 1 | | | | 1 | | 1 | | | 3 | 1 | 1 | | | | 1 | 1 | | 3 | | 7 | 1 | 2 | | | 12 | 23 |
| 15(11) | 1 | | 1 | | 1 | 1 | 3 | | | 6 | 1 | 2 | | 1 | 1 | | | 2 | 3 | | 9 | 4 | 1 | | | 15 | 37 |
| 16(12) | 1 | 2 | 1 | | 3 | 2 | 2 | 1 | 1 | 7 | 5 | 2 | 1 | | | 1 | 1 | 4 | 6 | | 9 | 4 | | 1 | | 19 | 55 |
| 17(13) | | | 1 | | 2 | | 4 | | 1 | 3 | 3 | 1 | | | 2 | 1 | | 1 | 4 | | 17 | 4 | 1 | 1 | | 14 | 46 |
| 18(14) | | 1 | 1 | | 2 | | 2 | | 3 | 3 | 1 | | | 1 | 1 | | 1 | 1 | 4 | 1 | 6 | 5 | | 1 | 1 | 17 | 35 |
| 19(16) | 1 | | 1 | 1 | 1 | | 2 | | 1 | 3 | 1 | 1 | 1 | | | | | 2 | 3 | 1 | 14 | 2 | 2 | | | 16 | 37 |
| 20(17) | | | | | 2 | | 1 | | 1 | 1 | | 1 | | | | | | 1 | 4 | | 13 | 6 | | | | 9 | 30 |
| 21(18) | | | | | 2 | | 3 | | 1 | 4 | 2 | 1 | | 1 | | | 1 | 1 | 4 | | 12 | | 1 | 1 | | 13 | 34 |
| 22(19a) | | | 1 | | 1 | | 2 | 1 | 2 | 1 | 1 | 2 | 1 | | 2 | | | 1 | 6 | | 10 | 3 | 1 | 1 | | 16 | 36 |
| Total Patients | 10 | 5 | 9 | 2 | 17 | 10 | 22 | 3 | 15 | 22 | 16 | 15 | 4 | 5 | 9 | 4 | 6 | 14 | 21 | 5 | 22 | 18 | 12 | 10 | 5 | | 641 |

a. *NRAS*; b. *ALK*; c. *IDH1*; d. *ERBB4*; e. *VHL*; f. *MLH1*; g. *PIK3CA*; h. *CTNNB1*; i. *KIT*; j. *APC*; k. *BRAF*; l. *EGFR*; m. *FGFR1*; n. *RET*; o. *PTEN*; p. *FGFR2*; q. *ATM*; r. *KRAS*; s. *RB1*; t. *IDH2*; u. *TP53*; v. *SMAD4*; w. *STK11*; x. *SRC*; y. *GNAS*

**Table 2. Pathogenic mutation mapping of 5 genes.**

| Gene | n = 22 | | | | | | | | | | | | | | | | | | | | | |
|---|---|---|---|---|---|---|---|---|---|---|---|---|---|---|---|---|---|---|---|---|---|---|
| MLH1 | | * | * | | | | * | * | | * | * | * | * | | * | * | | | | | | |
| PIK3CA | + | + | + | + | + | + | + | + | + | + | + | + | + | + | + | + | + | + | + | + | + | + |
| APC | * | * | * | * | * | * | * | * | * | * | * | *+ | * | * | * | * | * | * | * | * | * | * |
| KRAS | +& | | | | | | *& | + | | | + | | & | + | + | +& | + | + | +& | & | + | + |
| TP53 | +% | +% | +* | + | +# | +* | +*# | +*% | +*% | + | +* | +* | +*# | +% | +*& | + | +*% | +*% | +* | +*& | +% | +*%# |
| Sample | 1 | 2 | 3 | 4 | 5 | 6 | 7 | 8 | 9 | 10 | 11 | 12 | 13 | 14 | 15 | 16 | 17 | 18 | 19 | 20 | 21 | 22 |

*Nonsense; +Missense; #Frameshift; %Splice-site; &Silent; None

**Table 3. Subjects with co-occurring mutation.**

| Co-occurring mutation | Number of subjects |
|---|---|
| APC + TP53 + PIK3CA + KRAS + MLH1 | 6 |
| APC + TP53 + PIK3CA + KRAS | 8 |
| APC + TP53 + PIK3CA + MLH1 | 4 |
| APC + TP53 + PIK3CA | 4 |
| **Total** | 22 |

## APC mutation

Gene mutation occurred in 100% of subjects with 17 SNVs (16 missense and 1 nonsense). Mutation cluster regions (MCR) were located in exon 14–17. Median of SNV frequency was 4 (range 1–10). The most frequently occurred SNV was Q879* (Table 4).

## KRAS mutation

In this study, KRAS mutation occurred in 14 of 22 patients (63,6%). Nine SNVs were identified in 3 types of mutations, i.e., missense, nonsense, and silent. The nonsense mutation causes termination of codon 22, missenses occurred in 6 codons, and silent in 2 codons. The most frequently occurred SNVs are T20 = in 4 subjects, A146T, and P34L in 3 subjects (Table 5).

## TP53 mutation

TP53 mutation also occurred in 100% subjects in with 65 SNVs categorized into 5 types of mutations i.e. (1) missense, (2) nonsense; (3) frameshift; (4) silent; (5) splice-site. In missense mutation, the two most frequent SNVs are M237I and C238Y. In nonsense mutation, the two most frequent SNVs are R342* and R213* (6 of 22 patients) (Tables 6 and 7).

**Table 4. APC mutation.**

| Nonsense mutation | | | Missense mutation | | |
|---|---|---|---|---|---|
| SNV (n = 16) | | | SNV (n = 1) | | |
| Nucleotide change | Codon | Number (n = 22) | Nucleotide change | Codon | Number (n = 22) |
| C>T | Q879* | 10 | C>T | T1493M | 1 |
| | Q1123* | 8 | | | |
| | R876* | 6 | | | |
| | R1114* | 6 | | | |
| | Q1367* | 6 | | | |
| | Q1517* | 5 | | | |
| | Q1095* | 4 | | | |
| | Q1303* | 4 | | | |
| | Q1096* | 4 | | | |
| | Q1378* | 4 | | | |
| | Q1291* | 2 | | | |
| | Q1294* | 2 | | | |
| | Q1429* | 2 | | | |
| | Q1444* | 1 | | | |
| | R1450* | 1 | | | |
| | Q1469* | 1 | | | |

**Table 5.** *KRAS* mutation.

| Nonsense mutation | | | Missense mutation | | | Silent mutation | | |
|---|---|---|---|---|---|---|---|---|
| SNV (n = 1) | | | SNV (n = 6) | | | SNV (n = 2) | | |
| Nucleotide change | Codon | Number (n = 14) | Nucleotide change | Codon | Number (n = 14) | Nucleotide change | Codon | Number (n = 14) |
| C>T | Q22* | 1 | G>A | A146T | 3 | G>A | T20 = | 4 |
| | | | | V14I | 2 | C>T | G13 = | 3 |
| | | | | G13S | 2 | | | |
| | | | | A59T | 1 | | | |
| | | | C>T | P34L | 3 | | | |
| | | | | T58I | 2 | | | |

## *PIK3CA* mutation

Mutation of PIK3CA occurred in exons 2, 5, 7, 8, 10, 19, and 21. In this study, *PIK3CA* missense mutations were identified in all subjects. Median of SNV frequency was 4 (range 1–16). The most frequently occurred SNV was G914R (Table 8).

**Table 6.** *TP53* mutation.

| Missense mutation | | | | | |
|---|---|---|---|---|---|
| SNV (n = 49) | | | | | |
| Nucleotide change | Codon | Number (n = 22) | Nucleotide change | Codon | Number (n = 22) |
| G>A | M237I | 8 | A>G | M237V | 2 |
| | C238Y | 7 | | H214R | 2 |
| | R248Q | 6 | | K132E | 1 |
| | C277Y | 6 | | Q192R | 1 |
| | G245S | 6 | | N235D | 1 |
| | G245D | 5 | | Y236C | 1 |
| | G244D | 4 | C>T | S127F | 6 |
| | V197M | 4 | | R248W | 6 |
| | R213Q | 4 | | R282W | 5 |
| | R175H | 4 | | T256I | 4 |
| | E258K | 4 | | A138V | 4 |
| | R273H | 4 | | P152L | 4 |
| | R196Q | 3 | | L194F | 4 |
| | C135Y | 3 | | P250L | 3 |
| | G154S | 3 | | R273C | 3 |
| | R280K | 3 | | P152S | 2 |
| | R267Q | 2 | | T155I | 2 |
| | E285K | 2 | | P278L | 2 |
| | E286K | 1 | | R175C | 1 |
| | R290H | 1 | G>C | V272L | 1 |
| | C275Y | 1 | T>C | L755S | 1 |
| | G266E | 1 | | F134L | 1 |
| | R249K | 1 | | C238R | 1 |
| | R156H | 1 | | L252P | 1 |
| | R158H | 1 | | | |

**Table 7. *TP53* mutation (cont.).**

| Nonsense mutation | | | Frameshift mutation | | | Splice-site mutation | | | Silent mutation | | |
|---|---|---|---|---|---|---|---|---|---|---|---|
| SNV (n = 6) | | | SNV (n = 1) | | | SNV (n = 6) | | | SNV (n = 3) | | |
| Nucleotide change | Codon | Number (n = 22) | Nucleotide change | Codon | Number (n = 22) | Nucleotide change | Codon | Number (n = 22) | Nucleotide change | Codon | Number (n = 22) |
| C>T | R342* | 6 | G>A | M1? | 4 | c.919+1G>A | p.? | 4 | C>T | G244 = | 1 |
| | R213* | 6 | | | | c.673-1G>A | p.? | 3 | G>A | V272 = | 1 |
| | R196* | 5 | | | | c.560-1G>A | p.? | 3 | C>A | R213 = | 1 |
| | Q136* | 4 | | | | c.994-1G>A | p.? | 2 | | | |
| | Q165* | 1 | | | | c.559+2T>C | p.? | 1 | | | |
| G>A | W91* | 2 | | | | c.376-2A>G | p.? | 1 | | | |

## *MLH1* mutation

*MLH1* mutation occurred in 10 of 22 (45,45%) subjects. The nonsense mutation occurred in exon 9–13, causing termination in 4 codons. The most frequently occurred SNV was Q391* (Table 9).

## Biological behavior of malignancy with co-occurring mutation (Table 10)

Co-occurring mutations of *APC*, *TP53*, *PIK3CA*, and *KRAS* were identified in 8 patients with an average age of 48,5 years old, with locally-advanced stage (n = 5), located in the rectum (n = 6), well-differentiated (n = 6), and positive lymphovascular invasion (n = 5).

**Table 8. *PIK3CA* mutation.**

| Missense mutation | | |
|---|---|---|
| SNV (n = 9) | | |
| Nucleotide change | Codon | Number (n = 22) |
| G>A | G914R | 16 |
| | V71I | 7 |
| | R88Q | 5 |
| | G1049S | 4 |
| | R398H | 2 |
| | E542K | 1 |
| C>T | H1047Y | 3 |
| A>G | H1047R | 2 |
| T>A | N345K | 1 |

**Table 9. *MLH1* mutation.**

| Nonsense Mutation | | |
|---|---|---|
| SNV (n = 4) | | |
| Nucleotide change | Codon | Number (n = 10) |
| C>T | Q391* | 6 |
| | Q382* | 5 |
| | Q409* | 2 |
| | Q398* | 1 |

**Table 10. Clinical manifestation of each combination of co-occurring mutations.**

| | *APC + TP53 + PIK3CA + KRAS* (Cluster 1) | *APC + TP53 + PIK3CA + MLH1* (Cluster 2) | *APC + TP53 + PIK3CA + KRAS + MLH1* (Cluster 3) | *APC + TP53 + PIK3CA* (Cluster 4) |
|---|---|---|---|---|
| | **n = 8** | **n = 4** | **n = 6** | **n = 4** |
| **Age (*mean*) (range)** | *Mean 48,5±16 (27–75)* | *Mean 52,3±19 (27–67)* | *Mean 58,7±13 (40–74)* | *Mean 56,3±22 (39–87)* |
| ≥50 y.o. | 4 | 3 | 4 | 2 |
| <50 y.o. | 4 | 1 | 2 | 2 |
| **Stage** | | | | |
| Early | 1 | 0 | 1 | 0 |
| Locally-advanced | **5** | 3 | 4 | 3 |
| Advanced | 2 | 1 | 1 | 1 |
| **Lymphovascular invasion** | | | | |
| Yes | **5** | 0 | 4 | 4 |
| No | 3 | 4 | 2 | 0 |
| **Tumor location** | | | | |
| Group 1 | 1 | 1 | 0 | 1 |
| Group 2 | 1 | 0 | 3 | 0 |
| Group 3 | **6** | 3 | 3 | 3 |
| **Grade** | | | | |
| Well | **6** | 2 | 5 | 0 |
| Moderate | 1 | 0 | 0 | 2 |
| Poor | 1 | 2 | 1 | 2 |
| **Mortality** | | | | |
| Yes | 4 | 1 | 3 | 3 |
| No | 4 | 3 | 3 | 1 |

Co-occurring mutations of *APC*, *TP53*, *PIK3CA*, and *MLH1* were identified in 4 patients, with an average age of 52,3 years old, with locally advanced stage (n = 3), located in the rectum (n = 3), without lymphovascular invasion.

Quintuple mutations were identified in 6 patients, dominated by older age, locally-advanced stage, well-differentiated, positive lymphovascular invasion, and located in the rectum or left colon.

## Survival

Patients with co-occurring mutations of *APC*, *TP53*, *PIK3CA*, and *MLH1* (cluster 2) had the longest median life expectancy (1197 days) compared to cluster 1 with the shortest median life expectancy (577 days) (Table 3, Fig 1).

Fifty percent of subjects of cluster 1 and 3 were deceased in less than six months after therapy; in cluster 4, 50% of subjects were deceased before month 15. Cluster 2 can survive up to 30 months after therapy and only 1 patient deceased afterward. Cluster 1 and 4 show the highest mortality rate with the highest number of deceased patients in the shortest period compared to other clusters (Fig 2).

## Other findings

Early recurrence (<5 years) occurred in 2 patients of cluster 4, of which 1 patient underwent neoadjuvant chemoradiation and adjuvant chemotherapy (MFOLFOX6), and another was given XELOX after surgery. Both patients have a disease-free interval of 15 months.

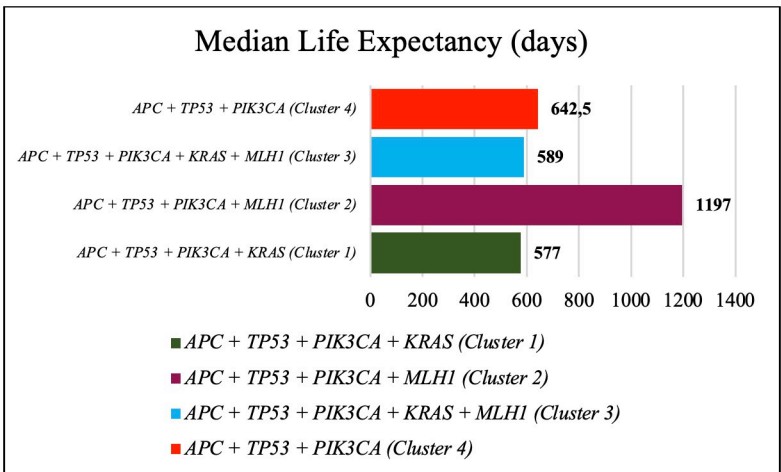

**Fig 1. Median life expectancy in days.**

One patient was given anti-*EGFR* therapy (cetuximab) + MFOLFOX6. The patient's PCR result for *KRAS* was wild-type. There is no therapeutic response data due to the patient's death during midcycle (127 days after surgery). This patient was included in cluster 1 (with *KRAS* mutation) and had *EGFR* mutation (rs121913467).

One patient was given anti-*VEGF* therapy (bevacizumab) + MFOLFOX6 after being diagnosed with local recurrence after 1-year of oral capecitabine and had a complete response to bevacizumab. This patient was included in cluster 2 with noted *BRAF* mutation (rs121913353).

Two of 22 patients had a family history of malignancy (Table 11). Germline mutation of *STK11* was identified in one patient with a family history of colon cancer. Meanwhile, two germline mutations of *TP53* were identified in another patient with a family history of breast cancer.

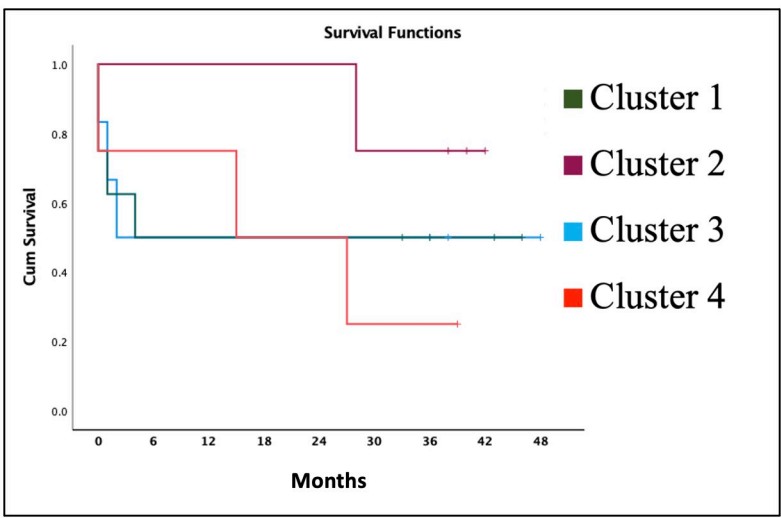

**Fig 2. Survival rate based on co-occurring mutation.**

**Table 11. Patients characteristics.**

| Variables | | Numbers | % |
|---|---|---|---|
| Age | < 50 y.o. | 9 | 41 |
| | ≥50 y.o. | 13 | 59 |
| Gender | Male | 13 | 59 |
| | Female | 9 | 41 |
| Death | Yes | 11 | 50 |
| | No | 11 | 50 |
| Grade | Well | 13 | 59 |
| | Moderate | 6 | 27 |
| | Poor | 3 | 14 |
| Stage | 1 | 2 | 9,1 |
| | 2A/B/C | 5/1/- | 22,7/4,5/- |
| | 3A/B/C | -/9/- | -/40,9/- |
| | 4A/B/C | 2/1/2 | 9,1/4,5/9,1 |
| Lymphovascular invasion | Yes | 13 | 59 |
| | No | 9 | 41 |
| Tumor location | Group 1 | 3 | 14 |
| | Group 2 | 4 | 18 |
| | Group 3 | 15 | 68 |
| Perioperative management | None | 13 | 59 |
| | Neoadjuvant chemoradiation | - | - |
| | Adjuvant chemotherapy | 7 | 32 |
| | Neoadjuvant chemoradiation + adjuvant chemotherapy | 2 | 9 |
| Family history of cancer | Colon cancer | 1 | 9 |
| | Breast cancer | 1 | |
| Average interval from diagnosed to death | | 259 days (3–882) | |

Group 1: Caecum to two-thirds proximal of the transverse colon; Group 2: One-third distal of the transverse colon to sigmoid; Group 3: Rectum to anus.

## Discussion

Colorectal cancer (CRC) patients in Indonesia are dominated by males (59%), more than 50 years old (59%), with well-differentiated (59%), stage 3B (40,9%), located in the rectum (68%). Recently, the incidence of CRC in young adults increased by 1,4% per year, influenced by obesity and a sedentary lifestyle [13]. High percentage of the locally-advanced stage on hospital admission can be caused by low educational level about CRC risk factors and importance of screening, especially in individuals with a family history of malignancy. The intricate system of national health insurance also has a role in slacking patients with unspecific complaints to see doctors before having an apparent disorder and getting worse. These are several reasons that cause a delay in the diagnosis and management of CRC.

The heterogeneous and dynamic nature of the CRC is related to its overlapping pathways of carcinogenesis. There are four principles of neoplasia in CRC, (1) colorectal tumors arise due to the activation of proto-oncogene mutations into oncogenes and inactivation of tumor suppressor genes [14]; (2) at least mutations in any 4–5 genes are required for malignant formation; (3) accumulation of numbers is more important than the sequence of mutations in determining tumor biologic behavior; (4) the mutated tumor suppressor gene continues to express the phenotype without loss of heterozygosity [2].

The theory of colorectal neoplasia, namely adenoma-carcinoma sequence (ACS), states that the presence of an adenoma must precede the formation of colorectal carcinoma [1, 2]. Mutations in the tumor suppressor gene, *APC*, triggered changes in the normal intestinal mucosal epithelium to adenoma. It can be detected in the aberrant crypt foci (ACF), a precursor lesion that occurred early in the beginning of the formation of adenomatous polyps and can only appear in dysplastic lesions [15].

All subjects (100%) in this study had nonsynonymous mutations in *APC*. Only two patients had adenomas on colonoscopy. One of those had tubulous adenomas with mild dysplasia on colonoscopy and a first-degree relative with CRC. Nonsense mutated *APC* was found at codons 879, 1095, 1123, which completely stopped glutamine production (Q). Meanwhile, in another patient with villous adenomas and well-differentiated adenocarcinoma, nonsense mutations were found at codons 876, 879, 1096, 1291, 1294, and 1517 that stopped the production of the amino acids glutamine (Q) and arginine (R). Mutations in *APC* have high-penetrance that can reach 100% for FAP and CRC [16–19]. In contrast to the Japanese population, whose *APC* mutations scattered at codons 142–1513, subjects in this study had *APC* mutations occur at codons 876–1517 with mutation cluster regions (MCR) in exons 14–17 [20, 21].

After the normal mucosal epithelium turned into an early adenoma, *KRAS* mutation occurred subsequently triggering early to intermediate adenoma. In contrast to *APC*, *KRAS* can act on nondysplastic ACF precursor lesions [15].

In this study, mutations in the *KRAS* gene occurred in 14 of 22 samples (63.6%) at 9 codons and were most commonly found in the older age group, locally-advanced stage, well-differentiated/low grade, with positive lymphovascular invasion, and located at the rectum. There were differences in codon location in missense mutation between Jakarta (Indonesia) and the United States population, i.e., codons 13, 14, 34, 58, 59, 146 VS 12, 13, 61, 146 [22]. In addition, nonsense mutations were also found at codon 22 which only occurred in 1 patient. This patient was diagnosed with stage 2A (pT3N0M0) CRC undergoing elective curative resection and 8 cycles of capecitabine adjuvant chemotherapy with complete response. Mutation located in codon 12 has more aggressive behavior than codon 13 because patients were commonly presented in advanced stage [22]. Nevertheless, several cases with metastases involving *KRAS* mutation in this study were found in 3 of 5 samples without the involvement of codon 12.

*KRAS* mutation can occur concomitantly with *APC* mutation leading to increased accumulation of β-catenin in the cytoplasm by destroying its binding to E-cadherin, which increased due to loss of mutated *APC* degradation function. This causes the Wnt signal to become more active so that motility and cell invasion are more aggressive than normal [15, 18, 21, 23–26]. In CRC, the combination of *APC* and *KRAS* mutations (co-occurring mutations) can occur up to 80%, whereas it only occurred in 63.6% of subjects in this study [27].

In this study, patients with *APC*, *TP53*, and *KRAS* mutations were predominantly ≥50 years old, with locally-advanced stage and positive lymphovascular invasion. Two shortest median life expectancy were found in patients with *KRAS* mutation (Fig 1); in addition, 50% of patients died within six months after therapy (Fig 2).

Before turning into carcinoma, intermediate adenomas differentiate into late adenomas triggered by mutations in the *SMAD4*, *CDC4*, and *DCC* genes [2, 7]. In this study, we found *SMAD4* nonsense and missense mutations in 18 of 22 patients (82%).

In ACS theory, late adenomas which developed into carcinomas have mutations in *TP53*, *TGFBR2*, *BAX*, and *IGF2R*. Mutated *TP53* was found in all subjects in this study in the form of nonsense, missense, frameshift, splice-site, and silent mutation. This study's five most frequently occurred codon locations were 237, 238, 127, G245S, and R248Q. Those are different compared to the world database in The Cancer Genome Atlas Program (TCGA) portal, which

stated that the five codon positions with the highest frequency were 175, 282, 248, R273H, and R273C [28].

In contrast to the UK population, in 64% (14 out of 22) subjects, *TP53* and *KRAS* mutations co-occurred [18, 21]. In Indian population, these two combinations were only found in 13 of 112 cases, whereas the study by Timar can occur in up to ~40% [27, 29]. *TP53* and *KRAS* activate different carcinogenesis pathways so that they rarely coexist [30].

Similar to *APC* and *TP53*, *PIK3CA* mutations were found in all subjects (100%) with 9 SNVs. *PIK3CA* has no role in the aggressive behavior of CRC, yet, when it occurs concurrently with *KRAS* mutations, evident aggressive behavior will be apparent, especially when it involves exons 9 or 20 or both [31, 32]. In this study, though mutations occurred in exons 2, 3, and 4, aggressive behavior presenting as locally-advanced stage and positive lymphovascular invasion can be found.

Mutations in *MLH1* can also occur in non-hereditary/sporadic CRC. The existence of microsatellite instability due to mutations in genes that play roles in the MMR system, such as *MLH1*, actually provides a good prognosis with a higher survival rate [33]. In this study, the group of cases with *MLH1* mutations alone had the highest median life expectancy and had a 30-month survival rate of up to 100%.

Referring to the colorectal neoplasia principle mentioned above, all subjects in this study involved activation of oncogenes (*PIK3CA* and *KRAS*) and inactivation of tumor suppressor genes (*APC*, *TP53*, and *MLH1*) and also involved a range of 8–19 mutated genes per person. In this study, mutated *APC* and *KRAS*, which are supposed to occur in the early sequence of ACS, supports what Fearon stated about the importance of mutational sequence in determining tumor biologic behavior [1, 2].

We are intensely aware of our study's limitations regarding small size of samples. Further research is genuinely required to complete the Indonesian profile mapping of colorectal cancer, especially in investigating our unique findings in each of the genes described and the relationship with ethnicities, diets, and lifestyles. This study is also applicable to other type of cancer in Indonesia population.

Nevertheless, this is the first study that fully describes the nature and location of five pathogenic mutated genes of CRC in the Indonesian population with its unique characteristics. Our population is compiled of various ethnicities with diverse diets and lifestyles which may have roles in contributing natures of the Indonesian version of CRC presented in locally-advanced stage with large tumor size and moderate-severe malnutrition status. This study is also the first in the world to examine the co-occurring mutations of *APC*, *TP53*, *PIK3CA*, *KRAS*, and *MLH1*.

## Conclusions

1. Different profile of pathogenic mutation in colorectal cancer patients is found in the Indonesian population

2. Mutated *APC*, *TP53*, and *PIK3CA* occurred in 100% of subjects, while *KRAS* and *MLH1* occurred in 63,6% and 45,4% of subjects

3. The longest median life expectancy occurred in the group of patients with mutations *APC*, *TP53*, *PIK3CA*, and *MLH1* with a 30-month postoperative survival of 100%.

4. The shortest median life expectancy occurred in the group of patients with *APC*, *TP53*, *PIK3CA*, and *KRAS* mutations with a 50% life expectancy <6 months post-treatment.

## Acknowledgments

The authors are grateful for a research grant (PUTI Q1) from Universitas Indonesia for founding the research with contract number: NKB-1312/UN2.RST/HKP.05.00/2020.

**Disclaimers:** The views expressed in the submitted article are authors' own and not an official position of the institution.

## Author Contributions

**Conceptualization:** Vania Myralda Giamour Marbun, Linda Erlina, Toar Jean Maurice Lalisang.

**Data curation:** Linda Erlina.

**Formal analysis:** Linda Erlina.

**Investigation:** Vania Myralda Giamour Marbun.

**Methodology:** Vania Myralda Giamour Marbun, Toar Jean Maurice Lalisang.

**Supervision:** Toar Jean Maurice Lalisang.

**Writing – original draft:** Vania Myralda Giamour Marbun.

**Writing – review & editing:** Vania Myralda Giamour Marbun.

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
