## [Decision Letter · Decision Letter 0]

25 Feb 2022

PONE-D-22-00968Indonesian Genomic Landscape of Pathogenic Mutation of APC, KRAS, TP53, PIK3CA, and MLH1 in Colorectal Cancer – How Different We Are from the WorldPLOS ONE

Dear Dr. Vania,

Thank you for submitting your manuscript to PLOS ONE. After careful consideration, we feel that it has merit but does not fully meet PLOS ONE’s publication criteria as it currently stands. Therefore, we invite you to submit a revised version of the manuscript that addresses the points raised during the review process.

We look forward to receiving your revised manuscript.

Kind regards,

Avaniyapuram Kannan Murugan, M.Phil., Ph.D.

Academic Editor

PLOS ONE

Journal Requirements:

Additional Editor Comments :

The manuscript received a positive response from the reviewers and needs to be improved prior to publication per the reviewers comments.

Reviewers' comments:

Reviewer's Responses to Questions

**Comments to the Author**

1. Is the manuscript technically sound, and do the data support the conclusions?

Reviewer #1: Yes

Reviewer #2: Yes

2. Has the statistical analysis been performed appropriately and rigorously? 

Reviewer #1: Yes

Reviewer #2: Yes

3. Have the authors made all data underlying the findings in their manuscript fully available?

Reviewer #1: Yes

Reviewer #2: Yes

4. Is the manuscript presented in an intelligible fashion and written in standard English?

Reviewer #1: Yes

Reviewer #2: Yes

5. Review Comments to the Author

Reviewer #1: The article is the first study to analyze genomic profile of colorectal cancer patients in Indonesian population. The authors utilized an advanced genomics technology and identified mutations. The article is well written, and the data are sufficient to support the conclusion.

Minor things

[1] Typo in page 18 line 12: “underwent neoadjuvar ~” should be “underwent neoadjuvant ~”

[2] What is “Months” in the same page 18 line 12?

Reviewer #2: The authors have herein provided a snapshot of the mutagenic properties of colorectal cancer in a population of patients in Indonesia from 2017-2018.

This clinical synopsis is well-developed and interesting. Further, it is potentially applicable to future studies involving other populations as well as different cancer types, as a number of the predominant pathways and genes indicated here have been shown to play a role in a variety of malignancies across multiple populations.

Minor Revisions:

Page 25:

"(2) at least mutations in 4-5 genes are required for malignant formation; " - Please make this clearer for the reader. Are there specific genes you could identify that would frequently be mutated in order to induce disease, or does this simply refer to 4-5 random mutations?

6. PLOS authors have the option to publish the peer review history of their article (what does this mean?). If published, this will include your full peer review and any attached files.

Reviewer #1: No

Reviewer #2: No

---

## [Author Response · Author response to Decision Letter 0]

5 Mar 2022

Reviewer #1: The article is the first study to analyze genomic profile of colorectal cancer patients in Indonesian population. The authors utilized an advanced genomics technology and identified mutations. The article is well written, and the data are sufficient to support the conclusion.

Minor things

[1] Typo in page 18 line 12: “underwent neoadjuvar ~” should be “underwent neoadjuvant ~”

[2] What is “Months” in the same page 18 line 12?

I already correct it to underwent neoadjuvant. The word “Months” is already deleted, and it was actually part of Figure 2. I already correct it and re upload it as well in Figure 2.

Reviewer #2: The authors have herein provided a snapshot of the mutagenic properties of colorectal cancer in a population of patients in Indonesia from 2017-2018.

This clinical synopsis is well-developed and interesting. Further, it is potentially applicable to future studies involving other populations as well as different cancer types, as a number of the predominant pathways and genes indicated here have been shown to play a role in a variety of malignancies across multiple populations.

Minor Revisions:

Page 25:

"(2) at least mutations in 4-5 genes are required for malignant formation; " - Please make this clearer for the reader. Are there specific genes you could identify that would frequently be mutated in order to induce disease, or does this simply refer to 4-5 random mutations?

 “at least mutations in 4-5 genes are required for malignant formation”

I cited this sentence from Reference no.2 (Fearon ER, Vogelstein B. A Genetic Model for Colorectal Tumorigenesis. Cell. 1990;61:759-767).

The study explained that malignancy requires median of four to five allelic losses per tumor.

In earlier years, according to adenoma-carcinoma sequence theory (1990) by Vogelstein, only certain genes are known to be associated in each phase, but nowadays, with more advanced technology, more genes (rather than described in 1990) are involved.

So it is implicitly random, but saying random in this sentence seem inappropriate.

I add the word “any” in "at least mutations in any 4-5 genes are required ……”

---

## [Decision Letter · Decision Letter 1]

29 Mar 2022

PONE-D-22-00968R1Indonesian Genomic Landscape of Pathogenic Mutation of APC, KRAS, TP53, PIK3CA, and MLH1 in Colorectal Cancer – How Different We Are from the WorldPLOS ONE

Dear Dr. Marbun,

Thank you for submitting your manuscript to PLOS ONE. After careful consideration, we feel that it has merit but does not fully meet PLOS ONE’s publication criteria as it currently stands. Therefore, we invite you to submit a revised version of the manuscript that addresses the points raised during the review process.

We look forward to receiving your revised manuscript.

Kind regards,

Avaniyapuram Kannan Murugan, M.Phil., Ph.D.

Academic Editor

PLOS ONE

Journal Requirements:

Additional Editor Comments:

1. Modify the original title "Indonesian Genomic Landscape of Pathogenic Mutation of APC, KRAS, TP53, PIK3CA, and MLH1 in Colorectal Cancer – How Different We Are from the World" to

"Genomic Landscape of Pathogenic Mutation of APC, KRAS, TP53, PIK3CA, and MLH1 in Indonesian Colorectal Cancer" 

2. Table 1: Stage 2A/B/C/      Number    5/1/-       Percentage should also follow similar format, i.e: 27/3/-.

3. All the gene names in: i) Title  ii) Table 2 iii) Table 3, 4, 9 iv) Figure legends and Table legends V) Manuscript text to be Italicized.

Reviewers' comments:

Reviewer's Responses to Questions

**Comments to the Author**

1. If the authors have adequately addressed your comments raised in a previous round of review and you feel that this manuscript is now acceptable for publication, you may indicate that here to bypass the “Comments to the Author” section, enter your conflict of interest statement in the “Confidential to Editor” section, and submit your "Accept" recommendation.

Reviewer #1: All comments have been addressed

2. Is the manuscript technically sound, and do the data support the conclusions?

Reviewer #1: Yes

3. Has the statistical analysis been performed appropriately and rigorously? 

Reviewer #1: Yes

4. Have the authors made all data underlying the findings in their manuscript fully available?

Reviewer #1: Yes

5. Is the manuscript presented in an intelligible fashion and written in standard English?

Reviewer #1: Yes

6. Review Comments to the Author

Reviewer #1: [1] Typo is corrected.

[2] Re-written Discussion section is fine.

All comments have been adequately addressed, and the manuscript is now ready for publication.

7. PLOS authors have the option to publish the peer review history of their article (what does this mean?). If published, this will include your full peer review and any attached files.

Reviewer #1: No

---

## [Author Response · Author response to Decision Letter 1]

30 Mar 2022

Response to Reviewers

Additional Editor Comments:

1. Modify the original title "Indonesian Genomic Landscape of Pathogenic Mutation of APC, KRAS, TP53, PIK3CA, and MLH1 in Colorectal Cancer – How Different We Are from the World" to

"Genomic Landscape of Pathogenic Mutation of APC, KRAS, TP53, PIK3CA, and MLH1 in Indonesian Colorectal Cancer" 

Already Changed – Thank you

2. Table 1: Stage 2A/B/C/ Number 5/1/- Percentage should also follow similar format, i.e: 27/3/-.

Already changed – Thank you

3. All the gene names in: i) Title ii) Table 2 iii) Table 3, 4, 9 iv) Figure legends and Table legends V) Manuscript text to be Italicized.

Already changed – Thank you

Including italicized ALL the gene names in the manuscript.

NB: but I did not highlight all the italicized gene

---

## [Editor Report · Decision Letter 2]

4 Apr 2022

Genomic Landscape of Pathogenic Mutation of APC, KRAS, TP53, PIK3CA, and MLH1 in Indonesian Colorectal Cancer

PONE-D-22-00968R2

Dear Dr. Marbun,

We’re pleased to inform you that your manuscript has been judged scientifically suitable for publication and will be formally accepted for publication once it meets all outstanding technical requirements.

Kind regards,

Avaniyapuram Kannan Murugan, M.Phil., Ph.D.

Academic Editor

PLOS ONE
---

## [Editor Report · Acceptance letter]

6 Apr 2022

PONE-D-22-00968R2 

Genomic Landscape of Pathogenic Mutation of *APC, KRAS, TP53, PIK3CA*, and *MLH1* in Indonesian Colorectal Cancer 

Dear Dr. Marbun:

I'm pleased to inform you that your manuscript has been deemed suitable for publication in PLOS ONE. Congratulations! Your manuscript is now with our production department. 

Kind regards, 

on behalf of

Dr. Avaniyapuram Kannan Murugan 

Academic Editor

PLOS ONE